# Urban air pollution and emergency department visits related to central nervous system diseases

**Anna O. Lukina[1], Brett Burstein[2,3], Mieczysław Szyszkowicz** **[1]\***

**1** Environmental Health Science and Research Bureau, Health Canada, Ottawa, Canada, **2** Division of Pediatric Emergency Medicine, Department of Pediatrics, Montreal Children's Hospital, McGill University Health Centre, Montreal, Quebec, Canada, **3** Department of Epidemiology, Biostatistics and Occupational Health, McGill University, Montreal, Quebec, Canada

\* mszyszkowicz@yahoo.ca

## Abstract

Ambient air pollution has been associated with adverse neurological health outcomes. Ambient pollutants are thought to trigger oxidative stress and inflammation to which vulnerable populations, such as elderly may be particularly susceptible. Our study investigated the possible association between concentrations of ambient air pollutants and the number of emergency department (ED) visits for nervous system disorders among people residing in a large Canadian city. A time-stratified case-crossover study design combining data from the National Ambulatory Care Reporting System (NACRS) and the National Air Pollution Surveillance (NAPS) between 2004 and 2015 was used. Two air quality health indices were considered in additional to specific pollutants, including carbon monoxide (CO), nitrogen dioxide ($NO_2$), sulfur dioxide ($SO_2$), ozone ($O_3$) and fine particulate matter ($PM_{2.5}$). Weather condition data were included in the models. ED visits with a discharge diagnosis were identified using ICD-10 codes (G00-G99). The analysis was stratified by sex and age, also by seasons. The associations were investigated in arrays organized as 18 strata and 15 time lags (in days) for each pollutant. Overall, 140,511 ED visits were included for the analysis. Most ED visits were related to episodic and paroxysmal diagnoses (G40-G47, 64%), with a majority of visits for migraines (G43, 39%). Among females, an increase of 0.1ppm ambient CO was associated with an increased risk of paroxysmal diagnoses at day 1 (RR = 1.019 (95%CI 1.004–1.033)), day 6 (1.024 (1.010–1.039)) and day 7 (1.022 (1.007–1.036). $PM_{2.5}$ and $SO_2$, and air quality indices were similarly associated with ED visits for episodic and paroxysmal disorders in days 6 and 7. Findings highlight that ambient air pollution is associated with an increased number of ED visits for nervous system disorders, particularly visits for paroxysmal diagnoses.

## 1. Introduction

Air pollution is the fifth-leading cause of mortality, accounting for nearly 9% of associated mortalities worldwide [1]. The study of the detrimental health effects of ambient air pollution is complex, as the environment contains a mixture of various particles and gases, including volatile organic compounds, particulate matter (PM), nitrogen oxides ($NO_x$), ground-level

---

---

**Competing interests:** The authors have declared that no competing interests exist.

**Abbreviations:** AD, Alzheimer's disease; AQHI, Air quality health index; CI, Confidence interval; CIHI, Canadian institute health information; CO, carbon monoxide; ED, Emergency department; ICD-10, International classification of diseases,10$^{th}$ Edition; IQR, Interquartile range; IRB, Institutional review board; NACRS, National ambulatory care reporting system; NAPS, National air pollution surveillance; NO$_2$, nitrogen dioxide; O$_3$, ozone (O$_3$H8 = ozone measured as 8 hour maximum average); PD, Parkinson's disease; PM, Particulate matter; ppb, parts per billion; ppm, parts per million; RR, Relative risk; SO$_2$, sulfur dioxide; WHO, World health organization.

ozone (O$_3$), sulfur dioxide (SO$_2$), carbon monoxide (CO) and others. Moreover, pathogenesis is also dependent on weather conditions, including humidity, wind speed, atmospheric pressure, precipitation, temperature, and seasons. Fine PM with a particle aerodynamic diameter of 2.5μm (PM$_{2.5}$) or smaller, NO$_2$, and O$_3$ are often studied because such pollutants pose a significant threat to human health [2]. Both NO$_2$ and PM$_{2.5}$ are emitted directly into the atmosphere from industrial sources (i.e., burning of fossil fuels, including for power generation), forest fires, and vehicle exhaust, including heavy duty and diesel engine exhaust, whereas ground-level O$_3$ is formed mostly by photochemical reaction [3].

Exposure to air pollutants may result in more visits to doctors or emergency department (ED), hospital admissions, frequent use of prescription medications, loss of productivity, and general changes to personal quality of life. In the last decade it was discovered that exposure to some major air pollutants may lead to various neurological outcomes, especially in older adults, including dementia, Alzheimer's disease (AD) and Parkinson's disease (PD) [4–6], headaches and migraines [7–9], epilepsy and seizures [10, 11], overall cognitive decline [12], and depression [13]. For example, long-term exposure to PM$_{2.5}$ for women over 70 years of age was associated with cognitive decline [14], as well as decreases in brain gray and white matter volume [15]. However, short-term exposure to ambient NO$_2$, PM$_{2.5}$, SO$_2$, and CO was associated with some neurological outcomes, including but not limited to depression [13], migraines [7], and epilepsy [10, 11, 16]. Two Canadian studies conducted in the largest and most populous provinces, Ontario and Quebec, found that when older Canadians are long-term exposed to ambient PM$_{2.5}$ and NO$_2$, and other traffic-related air pollutants (i.e., ultrafine particles and black carbon), over time they may develop dementia [6, 17]. In a study conducted in five major Canadian cities, exposure to ambient SO$_2$ and PM$_{2.5}$ was associated with more ED visits for headaches and migraine attacks, especially among females [7]. In China, the frequency of outpatient visits for epilepsy was higher for men exposed to ambient SO$_2$ and O$_3$, whereas women were more sensitive to NO$_2$ exposure [16]. However, in another Chinese study, acute exposure to ambient NO$_2$, CO, and PM$_{2.5}$ was associated with increased hospitalization for epilepsy [10]. A recent review of studies in Europe, Asia, and America highlighted the relationship between exposure to ambient air pollution and neurologic development among the young, as well as cognitive decline among the elderly [18].

Vulnerable populations, including elderly people, newborns and young children, pregnant women, as well as those with underlying health conditions, appear to be particularly sensitive to the deleterious health effects of ambient air pollution [14, 19–22].

Understanding the relationship between air pollutants and human health is essential to estimate economic impacts and the burden of health system resource utilization. To date, there is a paucity of information addressing the association of air pollution exposure with a wide range of disorders of the nervous system over the full life course. Previous studies have focused predominantly on specific vulnerable populations and health outcomes [19, 20, 23], or on particular pollutants [4, 14, 23–26]. The objective of this study was to comprehensively examine potential associations between concentrations of ambient air pollution and the number of ED visits for central and peripheral nervous system diseases in a large urban Canadian city.

## 2. Materials and methods

### 2.1 Studied population

Health data on the number of daily ED visits related to physician-diagnosed central and peripheral nervous system diseases were based on the *International Classification of Diseases 10$^{th}$ Revision* (ICD-10) codes G00-G99 (Chapter VI: "Diseases of the nervous system") [27], and were obtained from the National Ambulatory Care Reporting System (NACRS) database

[28] for the period of April 01, 2004 to December 31, 2015, inclusively (overall 4,292 days or 140 months). Briefly, the NACRS database is based on the Canadian Institute of Health Information (CIHI) reporting system in Canada, where the data are collected from hospitals and ambulatory care centres in the province of Ontario [29]. In 2016, the enumerated population of Toronto was 2,731,571 according to the Canadian Census Division 2016. The population density of this region is 4,334 people per square kilometer. The study population included all individuals from newborn to >60 years with a home addresses located in the area determined by the Census Division of Toronto. The primary outcome was the number of ED visits with a discharge diagnosis of all nervous system disorders (ICD-10 codes; G00-G99), and sub-analyzed according to diagnostic sub-categories, including "inflammatory diseases of the central nervous system" (G00-G09), "systemic atrophies primarily affecting the central nervous system" (G10-G14), "extrapyramidal and movement disorders" (G20-G26), "other degenerative diseases of the nervous system" (G30-G32), "demyelinating diseases of the central nervous system" (G35-G37), "episodic and paroxysmal disorders" (G40-G47), "nerve, nerve root and plexus disorders" (G50-G59), "polyneuropathies and other disorders of the peripheral nervous system" (G60-G64), "diseases of myoneural junction and muscle" (G70-G73), "cerebral palsy and other paralytic syndromes" (G80-G83), and "other disorders of the nervous system" (G90-G99).

## 2.2 Air pollutants and meteorological conditions data

Long-term air pollution data were obtained from the National Air Pollution Surveillance (NAPS) program, maintained by Environment and Climate Change Canada [30]. Hourly data from seven automated fixed-site monitoring stations with approximate maximum distance among them of 15 kilometers were averaged to estimate daily air pollution concentrations for the whole Census Division of Toronto. Five major ambient air pollutants were studied, which included CO, $NO_2$, $PM_{2.5}$, $O_3$, and $SO_2$. For each pollutant, daily measurements taken at hourly intervals were obtained. A 24-hour average was used for all air pollutants with the exception to $O_3$, which was calculated as a daily maximum 8-hour average. Additionally, the Air Quality Health Index (AQHI) was calculated, based on the composite of three pollutants, $NO_2$, $PM_{2.5}$, and $O_3$, measured as 24-hour averages in Toronto. The AQHI incorporates air pollutant concentrations and health risk estimations determined by mortality rates in large Canadian cities [31]. For the Canadian public, these values are rounded and shown as integer numbers on a scale (1–10 and 10+, and also expressed by colours from blue to brown) to demonstrate the risk related to ambient air quality, where lower-scale numbers represent lower health risks and higher-scale numbers represent higher health risks. The AQHI values are generated using the following formula:

$$AQHI = \frac{1000}{10.4} \times (e^{0.000537*O3} + e^{0.000871*NO2} + e^{0.000487*PM2.5} - 3),$$

where the coefficients for air pollutants were estimated using mortality risks [31]. In addition, another value of the index, AQHI-x, was calculated using the same three air pollutants as AQHI, but $O_3$ was on a maximum 8-hour average. This index has a stronger representation of $O_3$ than in the AQHI calculations, because ground-level $O_3$ tends to reach higher concentrations during daytime, especially on hot sunny days. The constructed indices were applied to investigate simultaneous effects of three air pollutants on human health.

To control for seasonal fluctuations in the concentrations of some air pollutants, data were analyzed by season, defined as cold (October to March) and warm (April to September) periods, classified according to the mean temperatures for each month recorded in Toronto [30].

To control for variations in weather [32, 33], daily relative humidity and ambient temperature data were collected from one weather monitoring station located at the Toronto International airport, details of which are described elsewhere [34], and the values were stratified as potential confounders.

## 2.3 Statistical analyses

A time-stratified case-crossover analytical design was used for all measured and unmeasured time-invariant factors and confounders, such as socioeconomic factors or comorbidities [35]. The analyzed data on associations between air pollutants concentrations and the number of ED visits for nervous system outcomes were organized as time-series values with a day as the time unit. These data consist of daily counts of ED visits, daily concentrations of air pollutants, and daily values of ambient temperature and relative humidity. A time-stratified approach was applied to cluster the data using a calendar hierarchical structure, where days were nested in day of weeks, then further nested in months and then in years [36]. Such clusters grouped as four or five days and were separated from each other. Controlling for time variables (i.e., any trends or fluctuations, etc.) was undertaken as described previously [37]. In the fitted statistical models, temperature and relative humidity were added in the form of natural splines of three degrees of freedom. The concentrations and their lagged values were assigned for the corresponding days. A two-tailed test at the 0.05 significance level was applied. The coefficients (slope, *Beta*) related to air pollutants and their standard errors (SE*Beta*) were estimated by the applied statistical models. Using these values (i.e., *Beta*, SE*Beta*), relative risks (RR) could be calculated. The statistical analyses were performed as conditional quasi-Poisson regression models [37, 38]. The numerical calculations were done employing R statistical software using the procedure for *Generalized Non-linear Models* (the package *GNM*) with the option "quasi-poisson" [39]. The realized statistical models have the following form: "ModelFit = gnm(EDVisits~AirPollutant + ns(RelativeHumidity,3) + ns(Temperature,3)". The options family = quasipoisson, eliminate = factor (Cluster) were included in the specification. The conditional Poisson model avoids estimating cluster parameters. It is conditional upon the total counts in each constructed cluster.

The conditional Poisson model is represented as a multinomial model and it is given by the formula

$$N_{i,C}|YN_{.,C} \sim Multinomial\left(\pi_i = \frac{exp\left\{\beta^T x_i\right\}}{\sum_{j\in C} exp\left\{\beta^T x_j\right\}}\right),$$

where $N_{i,C}$ is the number of events (counts) on the cluster and $N_{.,C} = \Sigma_i N_{i,C}$ is the sum of events in each cluster. The parameters related to cluster are eliminated by conditioning on the sum of events on each cluster [37]. Here i is a day when ED visits occurs, $\beta$ is a row vector of the coefficients, and T denotes transposition. A vector x contains variables: air pollutant concentrations and weather factors.

In total, 2,160 statistical models {15 (time lags expressed as days) x 18 (strata) x 8 (air pollutants and air quality health indices)} were applied. Strata covered patients' demographic characteristics, including age group (0–10, 11–60 and >60 years old) and sex group (all, males and females), as well as seasonality expressed dichotomously. The numerical results from all models are listed in the Supplementary Materials and at the online location https://github.com/szyszkowiczm/NERVEToronto. This location also contains histograms (air pollutants, temperature, and relative humidity) and the map of Toronto.

## 2.4 Research ethics

All datasets are publicly available and de-identified, as such this study was deemed exempt from review by the Health Canada Research Ethics Board.

## 3. Results

Between April 1, 2004 and December 31, 2015, there were a total of 140,511 ED visits related to diseases of the nervous system (all G00-G99 codes) (Table 1). Overall, a majority of ED visits were by females (59.5%), particularly in the age ranges of 11–60 years old and >60 years. A total of 89,708 visits (64% of all ED visits) for the nervous system diseases were for episodic and paroxysmal disorders (G40-G47), among which nearly 40% were related to migraine attacks (G43) (Table 2).

Table 3 summarizes the descriptive statistics on five major air pollutants, two air quality health indices and weather variables for the entire study period and by warm (April to September) and cold (October to March) season. During cold season, the mean ambient temperature was captured at 1.6˚C (-22.2–23.5˚C), while during warm season, the mean ambient temperature was 17.0˚C (-4.2–31.2˚C) (Table 2). The mean levels of some air pollutants, $NO_2$, $PM_{2.5}$, and $SO_2$, were slightly higher in warm season than in cold season; however, none of the annual mean values for the studied air pollutants was above the respective Canadian Ambient Air Quality Standards (CAAQS) [40]. Descriptive statistics for all major pollutants and weather factors are also available in S2 Table in S1 File.

Nervous system diseases (G00-G99): When examining ED data for all G00-G99 codes, a total of 115 consistent positive associations were found, especially for exposure to ambient $NO_2$ (24 positive associations) and ambient CO (25 positive associations) (S1 Fig in S1 File), with higher number of associations on lag days 0 and 7 with corresponding 27 associations and 33 associations, respectively (S2 Fig in S1 File). More ED visits were noted during the warm season (S3 Fig in S1 File).

Episodic and paroxysmal disorders (G40-G47): ED visits for episodic and paroxysmal disorders were more frequent at higher levels of CO and $PM_{2.5}$, with 29 and 27 positive associations, respectively (Fig 1). The majority of cases (39%) were due to all classes of migraines (G43), which was associated with higher ambient CO and $PM_{2.5}$ exposure, especially on lag days 6 (54 associations) and 7 (47 associations) (Fig 2) and during colder season for females

**Table 1. Descriptive statistics on the number of ED visits for diseases of the nervous system (ICD-10 codes: G00 – G99) collected in the city of Toronto between April 1, 2004 and December 31, 2015.**

|  | ED visits for nervous system diseases | % |
|---|---|---|
| All individuals | 140,511 | 100 |
| *Sex*: |  |  |
| Male (M) | 56,909 | 40.5 |
| Female (F) | 83,602 | 59.5 |
| *Ages (in years)*: |  |  |
| 0–10 | 2,314 (M) and 1,978 (F) | 53.9 (M) and 46.1 (F) |
| 11–60 | 36,177 (M) and 57,456 (F) | 38.6 (M) and 61.4 (F) |
| 60+ | 18,418 (M) and 24,168 (F) | 43.2 (M) and 56.8 (F) |
| *Seasons (months)*[a]: |  |  |
| Cold | 67,631 | 48.1 |
| Warm | 72,880 | 51.9 |

[a]—cold season (October-March), warm season (April-September), F- female, M- male.

**Table 2. Comparison of frequencies for each diagnostic sub-code (G00-G99).**

| Code | Meaning | ED visits (%) |
|---|---|---|
| G00-09 | Inflammatory diseases of the central nervous system | 1,913 (1.4) |
| G10-14 | Systemic atrophies primarily affecting the central nervous system | 472 (0.3) |
| G20-26 | Extrapyramidal and movement disorders | 4,075 (2.9) |
| G30-32 | Other degenerative diseases of the nervous system | 1,704 (1.2) |
| G35-37 | Demyelinating diseases of the central nervous system | 1,816 (1.3) |
| G40-47 | Episodic and paroxysmal disorders: | 89,708 (63.8): |
| | • Epilepsy/seizures (G40 and G41) | 18,670 (20.8) |
| | • Migraine (G43) | 34,864 (38.9) |
| | • Other headaches syndromes (G44) | 8,159 (9.1) |
| | • Cerebral ischaemic attacks and other related syndromes (G45) | 21,914 (24.4) |
| | • Sleep disorders (G47) | 6,101 (6.8) |
| G50-59 | Nerve, nerve root and plexus disorders | 30,607 (21.8) |
| G60-64 | Polyneuropathies and other disorders of the peripheral nervous system | 3,436 (2.5) |
| G70-73 | Diseases of myoneural junction and muscle | 889 (0.6) |
| G80-83 | Cerebral palsy and other paralytic syndromes | 1,244 (0.9) |
| G90-99 | Other disorders of the nervous system | 4,650 (3.3) |
| | **TOTAL** | 140,514 (100) |

especially (Fig 3). Ambient $SO_2$ exposure (20 positive associations) and the AQHI (27 positive associations) were found to be related to ED visits for episodic and paroxysmal disorders (Fig 1). The RR and associated 95% confidence intervals (CI) were then calculated based on the interquartile range (IQR), which was 0.1 ppm for CO. For episodic and paroxysmal disorders, the RR and associated 95%CIs were determined for females exposed to CO on lag day 1 (1.019 (1.004–1.033), lag day 6 (1.024 (1.010–1.039)), and lag day 7 (1.022 (1.007–1.036) (Fig 4).

There was no apparent relationship between seasonality, however, older age contributed to the exposure-effect association, especially among women ≥60 years old exposed to CO on the concurrent day (0 lag day) (Fig 4).

## 4. Discussion

### 4.1 Summary of findings

The present study found a significant association of air pollutants and ED visits for nervous system diseases. In particular, episodic and paroxysmal disorders were most frequent, and visits for migraines accounted for nearly 40% of all paroxysmal disorders. The results revealed that females were more prone to visit the ED after being acutely (lags of 0–1 or 6–7 days) exposed to ambient CO, $SO_2$, and $PM_{2.5}$ pollutants. There was a consistent effect of ambient CO exposure on the number of ED visits for all nervous system diseases and episodic and paroxysmal disorders.

### 4.2 Potential mechanisms underlying the identified associations

Previous studies have shown a positive association between ambient air pollution and recurring headaches and migraine attacks [7–9, 32, 33, 41], especially when individuals are exposed to CO [42–44], $PM_{2.5}$ [8, 41], and $SO_2$ [7, 9]. Headache pathophysiology, including migraines and vascular headaches, may be due to nociceptive stimuli, which may trigger changes in the vasodilation of the cranial blood vessels and/or sensory nerve fibers [45, 46], and air pollution

**Table 3. Descriptive data on ambient air pollutants and meteorological conditions on a daily basis, collected in Toronto, Canada between April 1, 2004 and December 31, 2015.**

| Factors (units) | Seasons[a] | | | | All months | | CAAQS [d] |
|---|---|---|---|---|---|---|---|
| | Cold | | Warm | | | | |
| **Air pollutants** | Mean | Min/Max[b] | Mean | Min/Max[b] | Mean | IQR[c] | |
| $PM_{2.5}$ (µg/m$^3$) | 8.5 | 0.9/35.6 | 9.2 | 0.1/65.5 | 8.9 | 6.5 | 8.8 |
| $NO_2$ (ppb) | 13.8 | 3.2/46.5 | 18.0 | 4.3/59.8 | 16.1 | 8.8 | 17.0 |
| $O_3$ (ppb) | 23.9 | 1.7/56.6 | 23.2 | 2.4/62.2 | 23.5 | 12.8 | N/A |
| $O_3H8$ (ppb) | 42.3 | 11.0/94.0 | 44.9 | 9.0/107.0 | 43.7 | 19.0 | 62.0 (8-hour) |
| $SO_2$ (ppb) | 0.9 | -0.5/5.3 | 1.8 | 0/ 12.0 | 1.4 | 1.2 | 5.0 |
| CO (ppm) | 0.3 | 0/0.7 | 0.3 | 0/1.1 | 0.3 | 0.1 | N/A |
| **Air quality indices** | | | | | | | |
| AQHI | 2.8 | 1.1/5.8 | 3.2 | 1.1/7.6 | 3.0 | 1.0 | N/A |
| AQHI-x | 4.0 | 1.6/8.0 | 4.7 | 1.7/10.3 | 4.4 | 1.5 | N/A |
| **Weather** | | | | | | | |
| Temperature (˚C) | 1.6 | -22.2/23.5 | 17.0 | -4.2/31.2 | 9.5 | 16.7 | - |
| Relative Humidity (%) | 72.7 | 31.7/98.8 | 68.8 | 35.4/96.7 | 70.7 | 14.3 | - |

Notes:

[a] - cold season (October to March) and warm season (April to September),

[b] - Min is minimum and Max is maximum,

[c] - IQR is difference between the third (75[th] percentile) and first (25[th] percentile) quartiles,

[d] - Canadian Ambient Air Quality Standards for annual values of $PM_{2.5}$, $NO_2$, and $SO_2$ (CAAQS) (available on https://www.ccme.ca/en/air-quality-report#slide-7) are from 2020 for comparison purposes solely,

N/A-not applicable.

is known to cause changes to vascular and neural activity [47, 48]. Ambient air pollutants may be related to systemic inflammation, oxidative stress, apoptosis [49, 50] or brain oxygenation and metabolism [45].

Results suggest a difference in response to air pollutants between males and females, and among various age groups. There were more ED encounters for CNS disorders by females (between 11 and 60 years of age compared to males, overall (Table 1) and for episodic and paroxysmal disorders (Fig 3)). Observed differences may be attributable to biological, physiological, and behavioral differences among sexes. Population-based studies suggest that air pollution is associated with headaches and migraine attacks, especially in women [7–9, 41, 51].

Previous environmental health studies showed stronger effects from acute air pollutants exposure on human health. In a European study with 12 volunteers of mean age of 24 years, nine volunteers developed prolonged headaches following CO inhalation during the two study days [51]. Additionally, significant increases in heart rate and facial skin blood flow were observed among the exposed volunteers [51]. In a cross-sectional study with 4,073 patients (mean age: 40 years), increased incidence of headaches was detected after CO poisoning, with further symptoms of nausea, dizziness, and shortness of breath [52]. In an American study, higher hospitalization rates and mortality due to CO poisoning were found, especially among older individuals of 75 years of age or above [53]. However, symptoms from prolonged CO exposure normally occurs when levels exceed 70 ppm [43], which is very high compared to levels found in the current study (0.3 ppm).

Several studies have demonstrated a negative impact of air pollution exposure on development and progression of neurodegenerative disorders, including PD [54] and multiple sclerosis [55, 56], as well as potential for violent behavior resulting in 3.13% increase in risk for

| Air Pollution: | AQHI | AQHIX | CO | NO2 | O3 | O3H8 | PM2.5 | SO2 | Total |
|---|---|---|---|---|---|---|---|---|---|
| All | 2 | 2 | 2 | 0 | 0 | 2 | 2 | 2 | 12 |
| Female | 3 | 2 | 3 | 2 | 0 | 0 | 3 | 2 | 15 |
| Male | 0 | 0 | 0 | 0 | 0 | 1 | 0 | 0 | 1 |
| Warm All | 2 | 1 | 3 | 2 | 0 | 1 | 2 | 1 | 12 |
| Warm Female | 2 | 2 | 3 | 2 | 0 | 0 | 3 | 1 | 13 |
| Warm Male | 1 | 1 | 1 | 0 | 0 | 1 | 2 | 0 | 6 |
| Cold All | 3 | 2 | 2 | 0 | 0 | 3 | 3 | 2 | 15 |
| Cold Female | 6 | 3 | 5 | 4 | 0 | 6 | 5 | 4 | 33 |
| Cold Male | 0 | 0 | 0 | 0 | 1 | 2 | 0 | 0 | 3 |
| Age 0-10 All | 0 | 0 | 0 | 0 | 0 | 0 | 0 | 0 | 0 |
| Age 0-10 Female | 0 | 0 | 0 | 0 | 0 | 0 | 0 | 0 | 0 |
| Age 0-10 Male | 1 | 1 | 0 | 2 | 0 | 0 | 2 | 1 | 7 |
| Age 11-60 All | 1 | 1 | 1 | 0 | 0 | 0 | 2 | 0 | 5 |
| Age 11-60 Female | 4 | 1 | 3 | 1 | 0 | 0 | 2 | 2 | 13 |
| Age 11-60 Male | 0 | 0 | 0 | 0 | 0 | 0 | 0 | 0 | 0 |
| Age 60+All | 1 | 2 | 3 | 1 | 0 | 1 | 0 | 2 | 10 |
| Age 60+ Female | 1 | 2 | 3 | 3 | 0 | 0 | 1 | 3 | 13 |
| Age 60+ Male | 0 | 0 | 0 | 0 | 1 | 2 | 0 | 0 | 3 |
| Total | 27 | 20 | 29 | 17 | 2 | 19 | 27 | 20 | 161 |

**Fig 1. Total frequencies of all associations: 18 strata (rows), five air pollutants and two air quality health indices (columns) between ambient air pollutants levels and the number of ED visits for episodic and paroxysmal disorders (G40-G47) in Toronto, Canada from April 1, 2004 to December 31, 2015.** For visual representation: 0 (green) colour represents others than positive statistically significant associations.

homicide and inflicted injury [57]. Such increase in ED visits was observed especially among males, children and Hispanics exposed to ambient CO during warm season in California [57]. Possible explanation can be because of combination effect of the air pollutant and heat stress [57]. Such results are in agreement with another Canadian study showing that even small increases in ambient CO levels may be related to an increases in substance abuse visits [58]. There might be a clear hormesis associated with CO exposure, as such higher concentrations and prolonged exposure may result in neuroinflammation and disruption to blood-brain barrier and transfer of oxygen to the brain [42, 45, 50], but very small quantities may be used for therapeutic benefit in some cases [59].

| Lag | 0 | 1 | 2 | 3 | 4 | 5 | 6 | 7 | 8 | 9 | 10 | 11 | 12 | 13 | 14 | Total |
|---|---|---|---|---|---|---|---|---|---|---|---|---|---|---|---|---|
| AQHI | 0 | 0 | 0 | 0 | 0 | 1 | 8 | 9 | 3 | 2 | 3 | 0 | 0 | 1 | 0 | 27 |
| AQHI-X | 0 | 0 | 0 | 0 | 0 | 2 | 6 | 8 | 0 | 1 | 1 | 0 | 1 | 1 | 0 | 20 |
| CO | 2 | 4 | 1 | 0 | 0 | 4 | 9 | 7 | 1 | 0 | 0 | 1 | 0 | 0 | 0 | 29 |
| $NO_2$ | 1 | 1 | 2 | 0 | 0 | 2 | 4 | 5 | 0 | 0 | 1 | 1 | 0 | 0 | 0 | 17 |
| $O_3$ | 0 | 0 | 0 | 0 | 0 | 0 | 2 | 0 | 0 | 0 | 0 | 0 | 0 | 0 | 0 | 2 |
| $O_3$H8 | 0 | 0 | 0 | 0 | 0 | 2 | 7 | 4 | 2 | 1 | 1 | 0 | 2 | 0 | 0 | 19 |
| $PM_{2.5}$ | 0 | 0 | 0 | 0 | 0 | 0 | 10 | 9 | 3 | 3 | 1 | 0 | 0 | 1 | 0 | 27 |
| $SO_2$ | 0 | 0 | 0 | 0 | 0 | 0 | 8 | 5 | 3 | 2 | 2 | 0 | 0 | 0 | 0 | 20 |
| Total | 3 | 5 | 3 | 0 | 0 | 11 | 54 | 47 | 12 | 9 | 9 | 2 | 3 | 3 | 0 | 161 |

**Fig 2. Total frequencies for all associations: Five air pollutants + two air quality health indices (rows) and 15 time lags of 0–14 days (columns), between exposure to urban air pollutants levels and the number of ED visits for episodic and paroxysmal disorders (G40-G47) in Toronto, Canada between April 1, 2004 and December 31, 2015.** For visual representation: 0 (green) colour represents others than positive statistically significant associations.

Exposure to other pollutants, such as $PM_{2.5}$ and $SO_2$ may also contribute to central nervous system disorders. Exposure to both pollutants may result in higher prevalence of multiple sclerosis [55, 56] and higher risk of progression of PD [54] and AD [4]. In a Canadian study, exposure to ambient $SO_2$ and $PM_{2.5}$ was associated with increased risks of ED visits for depression

| Lag | 0 | 1 | 2 | 3 | 4 | 5 | 6 | 7 | 8 | 9 | 10 | 11 | 12 | 13 | 14 | Total |
|---|---|---|---|---|---|---|---|---|---|---|---|---|---|---|---|---|
| All | 0 | 0 | 0 | 0 | 0 | 0 | 6 | 6 | 0 | 0 | 0 | 0 | 0 | 0 | 0 | 12 |
| Female | 0 | 1 | 0 | 0 | 0 | 0 | 6 | 6 | 0 | 2 | 0 | 0 | 0 | 0 | 0 | 15 |
| Male | 0 | 0 | 0 | 0 | 0 | 0 | 1 | 0 | 0 | 0 | 0 | 0 | 0 | 0 | 0 | 1 |
| Warm All | 1 | 0 | 0 | 0 | 0 | 0 | 4 | 6 | 1 | 0 | 0 | 0 | 0 | 0 | 0 | 12 |
| Warm Female | 0 | 1 | 0 | 0 | 0 | 0 | 4 | 5 | 0 | 0 | 0 | 0 | 0 | 3 | 0 | 13 |
| Warm Male | 0 | 0 | 0 | 0 | 0 | 0 | 1 | 4 | 0 | 0 | 0 | 1 | 0 | 0 | 0 | 6 |
| Cold All | 0 | 0 | 0 | 0 | 0 | 1 | 5 | 3 | 4 | 0 | 0 | 0 | 2 | 0 | 0 | 15 |
| Cold Female | 0 | 2 | 2 | 0 | 0 | 2 | 7 | 7 | 4 | 5 | 3 | 0 | 1 | 0 | 0 | 33 |
| Cold Male | 0 | 0 | 0 | 0 | 0 | 1 | 2 | 0 | 0 | 0 | 0 | 0 | 0 | 0 | 0 | 3 |
| Age 0-10 All | 0 | 0 | 0 | 0 | 0 | 0 | 0 | 0 | 0 | 0 | 0 | 0 | 0 | 0 | 0 | 0 |
| Age 0-10 Female | 0 | 0 | 0 | 0 | 0 | 0 | 0 | 0 | 0 | 0 | 0 | 0 | 0 | 0 | 0 | 0 |
| Age 0-10 Male | 0 | 0 | 0 | 0 | 0 | 0 | 0 | 0 | 2 | 1 | 3 | 1 | 0 | 0 | 0 | 7 |
| Age 11-60 All | 0 | 0 | 0 | 0 | 0 | 0 | 1 | 4 | 0 | 0 | 0 | 0 | 0 | 0 | 0 | 5 |
| Age 11-60 Female | 0 | 1 | 0 | 0 | 0 | 0 | 4 | 6 | 1 | 0 | 1 | 0 | 0 | 0 | 0 | 13 |
| Age 11-60 Male | 0 | 0 | 0 | 0 | 0 | 0 | 0 | 0 | 0 | 0 | 0 | 0 | 0 | 0 | 0 | 0 |
| Age 60+All | 1 | 0 | 0 | 0 | 0 | 3 | 5 | 0 | 0 | 0 | 1 | 0 | 0 | 0 | 0 | 10 |
| Age 60+ Female | 1 | 0 | 1 | 0 | 0 | 3 | 6 | 0 | 0 | 1 | 1 | 0 | 0 | 0 | 0 | 13 |
| Age 60+ Male | 0 | 0 | 0 | 0 | 0 | 1 | 2 | 0 | 0 | 0 | 0 | 0 | 0 | 0 | 0 | 3 |
| Total | 3 | 5 | 3 | 0 | 0 | 11 | 54 | 47 | 12 | 9 | 9 | 2 | 3 | 3 | 0 | 161 |

**Fig 3. Total frequencies of all associations for ambient air pollutants and the number of ED visits for related episodic and paroxysmal disorders (G40-G47).** Eighteen strata (classified by patients' sex and age, as well as cold vs. warm seasons) examined and arranged in the rows, and 15 lags (expressed as days) are arranged in columns. For visual representation: 0 (green) colour represents other than positive statistically significant associations.

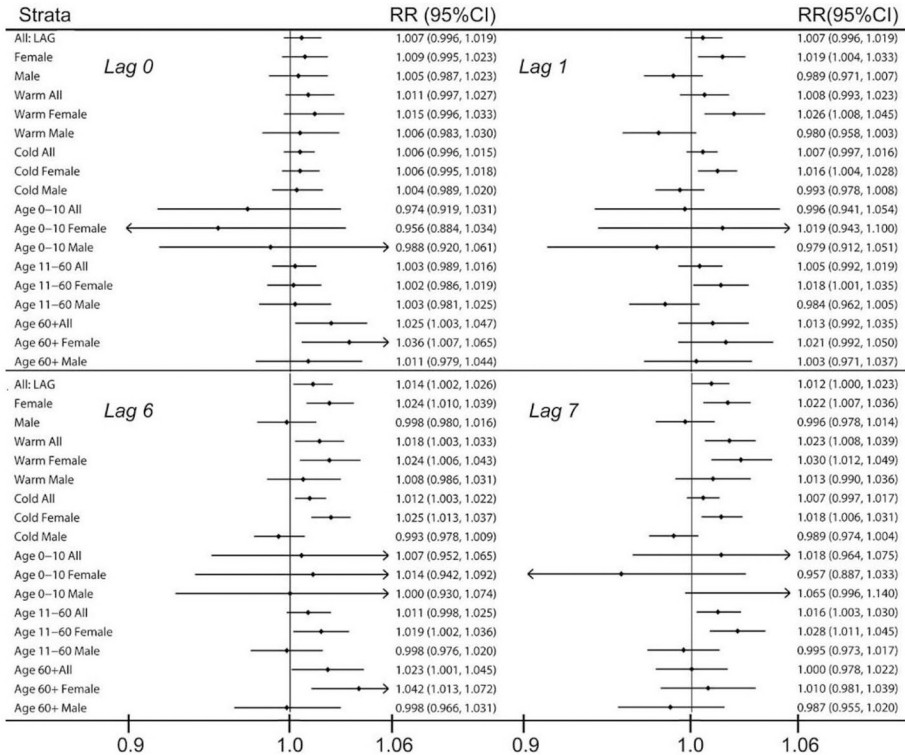

**Fig 4. Relative risks (RR) and 95% confidence intervals (95%CI) for an increase in a one interquartile range (CO, IQR = 0.1 ppm).** ED visits diagnosed with the ICD-10 codes G40-G47 for the entire study period in Toronto between 2004 and 2015.

[60]. Short-term exposure to ambient $SO_2$ resulted in increased 1.12 odds ratio of suicide risks, especially among males and well-educated individuals [61].

## 4.3 Strengths and limitations of the current study

The present study has several strengths. First, this study used ten years of longitudinal data on exposure and incorporated daily fluctuations in ambient air pollution concentrations into models estimating ED visits for nervous system diseases. To estimate the relative risks of a particular health outcome, the frequency of exposure to pollution prior to the health outcome is compared to control times (for example, a health outcome taken on a Monday is compared to those on other Mondays of the month). In addition, a case-crossover approach controls for all measured and unmeasured confounding factors, including socioeconomics and co-morbidities. The study considered the individual five major ambient air pollutants, as well as multi-pollutants expressed in AQHI (and AQHI-x) indices. The current study also considered both sexes across the lifespan (from newborn to ≥60 years of age) for patients attending the hospitals' ED.

There are some limitations to the present study. This study was unable to control for individuals' exposure duration, as there are disparities in individuals' lifestyles and habits. The study was also unable to control for personal exposure, including indoor and in-vehicle exposure, as well as proximity to major and local roads [62], as the current study was solely dependent on air pollution concentrations captured by seven fixed-site automated monitoring

stations. Secondly, it is possible that some admission data reported to the national NACRS database by facilities may be incomplete or inaccurate due to human error [29]. Additionally, previous patients' health information was not available, and thus, the study assessed the daily changes in air pollution concentrations and the number of ED visits for nervous system diseases in the given study period. Lastly, the multiplicity of comparisons among all five studied air pollutants, time lags of 14 days and 18 strata may introduce a possibility of erroneous associations (type 1 error). However, the study found consistent positive association between exposure to ambient air pollutants and nervous system diseases, as well as for the episodic and paroxysmal disorders alone.

## 5. Summary and future perspectives

In the present study, consistent positive exposure-effect associations were found between acute (i.e., concurrent day or a weeklong) exposure to ambient CO, $PM_{2.5}$ and $SO_2$ concentrations and more frequent utilization of ED for CNS diseases (i.e., episodic and paroxysmal disorders), and perhaps, attributable to migraine attacks among patients residing in such a large city as Toronto. Although the biological plausibility of the observed association between ambient air pollution and episodic and paroxysmal disorders, including headaches and migraine attacks, is still remaining unclear. Additionally, further studies should focus on other geographical regions with similar environmental exposures to either confirm or refute the consistency of the findings presented here.

## Supporting information

**S1 File.**
(DOCX)

## Acknowledgments

The authors acknowledge Environment and Climate Change Canada for providing the air pollution data from the National Air Pollution Surveillance (NAPS) network.

Parts of this material are based on data and information compiled and provided by the Canadian Institute for Health Information (CIHI). However, the analyses, conclusions, opinions and statements expressed herein are not necessarily those of CIHI.

## Author Contributions

**Conceptualization:** Mieczysław Szyszkowicz.

**Data curation:** Mieczysław Szyszkowicz.

**Formal analysis:** Anna O. Lukina, Brett Burstein.

**Investigation:** Mieczysław Szyszkowicz.

**Methodology:** Mieczysław Szyszkowicz.

**Project administration:** Mieczysław Szyszkowicz.

**Software:** Mieczysław Szyszkowicz.

**Supervision:** Mieczysław Szyszkowicz.

**Validation:** Anna O. Lukina, Brett Burstein.

**Visualization:** Anna O. Lukina.

Writing – **original draft:** Mieczysław Szyszkowicz.

Writing – **review & editing:** Anna O. Lukina, Brett Burstein.

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
