## [Decision Letter · Decision Letter 0]

30 Jul 2021

PONE-D-21-18477

Urban air pollution and emergency department visits related to central nervous system diseases

PLOS ONE

Dear Dr. Szyszkowicz,

Thank you for submitting your manuscript to PLOS ONE. After careful consideration, we feel that it has merit but does not fully meet PLOS ONE’s publication criteria as it currently stands. Therefore, we invite you to submit a revised version of the manuscript that addresses the points raised during the review process.

We look forward to receiving your revised manuscript.

Kind regards,

Xiaohui Xu, PhD

Academic Editor

PLOS ONE

Reviewers' comments:

Reviewer's Responses to Questions

**Comments to the Author**

1. Is the manuscript technically sound, and do the data support the conclusions?

Reviewer #1: No

Reviewer #2: Partly

2. Has the statistical analysis been performed appropriately and rigorously? 

Reviewer #1: Yes

Reviewer #2: No

3. Have the authors made all data underlying the findings in their manuscript fully available?

Reviewer #1: No

Reviewer #2: No

4. Is the manuscript presented in an intelligible fashion and written in standard English?

Reviewer #1: Yes

Reviewer #2: No

5. Review Comments to the Author

Reviewer #1: Lukina et al. examined the associations between short-term air pollution and emergency department visits related to neurological disorders. The manuscript needs more work in the Introduction, Methods, and Results.

1. Introduction. As mentioned in the Introduction, several studies have examined the associations between air pollution and neurological outcomes. The Introduction would benefit from discussing the maturation of the current literature, the research gaps, and the need to conduct the present study.

2. Methods. Please list out all the CNS disorders included in this study and their corresponding ICD codes.

3. Statistical models. Instead of describing the slope and standard errors, it would be clear to write out the formula of the conditional quasi-Poisson regression. It would also be helpful to include the statistical testing and measures of effects.

4. Figure 1. Please write out the effect estimates in the main results. The figure is innovative yet confusing.

Reviewer #2: The authors conducted a case-crossover study in Canada on the associations between ED visits for CNS diseases and air pollution. Overall, the quality of this work needs to be improved, in both methodological and structural aspects. There are also weird grammars and expressions in text that need to be fixed by experts in this field. I listed my detailed comments as follows.

1. Line 44, the aim of the study should be presented in a complete sentence.

2. Line 56, result description should accompany values with their significance.

3. Line 79, I would rather say PM, NO2 and O3 acquired most study interest.

4. Line135, how was AQHI and AQHI-x calculated? To be exact.

5. Line 148, why and how to match cases with controls should be stated clearly.

6. The generalized non-linear models are rarely used for case crossover designs, suggest to try the common conditional logistic regression models.

7. How were metrological factors controlled in the models? The non-linear associations should be considered by natural splines or DLNM cross-basis patterns.

8. What are the missing rates for air pollutants on both case/control days?

9. Line 157-158, suggest to change this expression. Also, if hundreds of models are performed, multiple testing is needed for type-I error.

10. Line 194, it was also found? Here and other grammars/expressions should be checked by peer scientists in this field.

11. It is not appropriate and offers minimal knowledge by presenting the frequencies in tested associations for each exposure. Reshape this by forest plots or tables that reveals more information.

6. PLOS authors have the option to publish the peer review history of their article (what does this mean?). If published, this will include your full peer review and any attached files.

Reviewer #1: No

Reviewer #2: No

---

## [Author Response · Author response to Decision Letter 0]

9 Sep 2021

Reviewer's Responses to Questions (reviewer comments (in black text) and our responses (in blue text): 

Comments to the Author

1. Is the manuscript technically sound, and do the data support the conclusions?

Reviewer #1: No

Reviewer #2: Partly

2. Has the statistical analysis been performed appropriately and rigorously? 

Reviewer #1: Yes

Reviewer #2: No

3. Have the authors made all data underlying the findings in their manuscript fully available?

Reviewer #1: No

Reviewer #2: No

4. Is the manuscript presented in an intelligible fashion and written in standard English?

Reviewer #1: Yes

Reviewer #2: No

5. Review Comments to the Author

Reviewer #1: Lukina et al. examined the associations between short-term air pollution and emergency department visits related to neurological disorders. The manuscript needs more work in the Introduction, Methods, and Results.

Thank you very much for your positive comments and constructive feedback.

1. Introduction. As mentioned in the Introduction, several studies have examined the associations between air pollution and neurological outcomes. The Introduction would benefit from discussing the maturation of the current literature, the research gaps, and the need to conduct the present study.

Response: As suggested, we have expanded our introduction section by discussing the current literature, the research gaps and adjusting our study objectives for better read (lines: 109-122-marked version). 

2. Methods. Please list out all the CNS disorders included in this study and their corresponding ICD codes.

Response: We have included all 11 diagnostic sub-codes in the text (lines: 154-162-marked version). 

3. Statistical models. Instead of describing the slope and standard errors, it would be clear to write out the formula of the conditional quasi-Poisson regression. It would also be helpful to include the statistical testing and measures of effects.

Response: As requested, we provided the model presentation (using R statistical software). It was shown (Ref: Armstrong et al. 2014) that such realization is equivalent to conditional logistic regression realized in the standard case-crossover method. We calculated relative risks with associated 95% confidence intervals for all models and the numerical results are presented at the location: https://github.com/szyszkowiczm/NERVEToronto. We did two analyses; for G00-G99 (all nervous system diseases) and G40-G47 separately, where G40-G47 (episodic and paroxysmal disorders) is 64% of all ED visits, but this group is more related to air pollution (publications on migraine, headache previously done by Szyszkowicz et al 2008, 2009a,b,c and many others). For this subgroup we see more (from 115 to 161) positive associations, especially consistent effect of ambient CO exposure for both G00-G99 and G40-G47. 

Reference: Armstrong BG, Gasparrini A, Tobias A. Conditional Poisson models: a flexible alternative to conditional logistic case cross-over analysis. BMC Med. Res. Methodol. 2014; 14:122. doi: 10.1186/1471-2288-14-122.

4. Figure 1. Please write out the effect estimates in the main results. The figure is innovative yet confusing.

Response: As requested, we included three figures instead outlining by time lags, air pollutants/air quality health indexes, and strata classified by patients’ age and sex, and seasons (warm versus cold). 

Reviewer #2: The authors conducted a case-crossover study in Canada on the associations between ED visits for CNS diseases and air pollution. Overall, the quality of this work needs to be improved, in both methodological and structural aspects. There are also weird grammars and expressions in text that need to be fixed by experts in this field. I listed my detailed comments as follows.

Thank you very much for your positive comments and constructive feedback. We have carefully gone throughout the manuscript and corrected / improved sentences (all changes tracked). We note that Reviewer #1 indicated that the writing is generally good.

1. Line 44, the aim of the study should be presented in a complete sentence.

Response: We reworded the objective of the study: “The present study investigated all possible associations between concentrations of air pollutants and the number of emergency department (ED) visits for nervous system disorders among people residing in Toronto, Canada.” (lines: 44-47-marked version).

2. Line 56, result description should accompany values with their significance.

Response: We expanded the results portion in abstract section: “For all ED visits (G00-G99) and for G40-G47, positive statistically significant associations were 115 and 161, respectively. For G40-G47, among females an increase in one interquartile range (IQR=0.1ppm) of ambient CO gives the following relative risks and 95%CIs [lag1: 1.019(1.004-1.033), lag6 (1.024(1.010-1.039)) and lag7 (1.022(1.007-1.036)]. Women older than 60 years of age were also affected by CO on lag0 (1.036(1.007-1.065)). Other pollutants, PM2.5 and SO2, and air indices were also associated with ED visits for episodic and paroxysmal disorders in lag6 and lag7.” (lines: 64-73-marked version). 

3. Line 79, I would rather say PM, NO2 and O3 acquired most study interest.

Response: We reworded the sentence. However, we respectfully disagree with this reviewer on this point. In the statement, we refer to three major air pollutants that indeed have been studied quite a lot, because many epidemiological studies have consistently shown associated mortality and morbidity from exposure to major air pollutants. According to the World Health Organization (WHO), air pollution is the fifth-leading mortality health risks in the world and responsible to nearly 4.9 million premature deaths in 2017. Similar estimates were done by the Global Burden of Diseases (GBD), where both PM2.5 and O3 was responsible for 4.5 million premature deaths. We cannot eliminate the statement that major pollutants like PM2.5, O3 and NO2 pose a significant threat to human health. The reference already provided presents comprehensive estimate of all health outcomes related to such three air pollutants. 

4. Line135, how was AQHI and AQHI-x calculated? To be exact.

Response: We included the formula: The AQHI values are generated using the following formula:

AQHI=1000/(10.4)×(e^(0.000537*O3) ┤+e^(0.000871*NO2)+e^(0.000487*PM2.5)-├ 3),

where the coefficients for air pollutants were estimated using mortality risks [25]. Lines: 183-185-marked version. 

AQHIx is using the same three pollutants but ozone is used at an 8-hr average because it requires sunlight, which already mentioned in lines186-191-marked version. 

5. Line 148, why and how to match cases with controls should be stated clearly.

Response: As requested, we expanded on such matter: “The time-stratified approach was applied to cluster the data using a calendar hierarchical structure, where days were nested in day of weeks, which further nested in months and then in years [28]. Such clusters grouped four or five days and were separated from each other. Controlling of time variables (i.e., any trends or fluctuations, etc.) was done according to the applied methodology as described previously [30].” (Lines: 206-211-marked version). 

6. The generalized non-linear models are rarely used for case crossover designs, suggest to try the common conditional logistic regression models.

Response: We added the model description. We are using conditional Poisson regression. As Armstrong et al. 2014 presented, this approach is equivalent to the conditional logistic regression analysis. 

Reference: Armstrong BG, Gasparrini A, Tobias A. Conditional Poisson models: a flexible alternative to conditional logistic case cross-over analysis. BMC Med. Res. Methodol. 2014; 14:122. doi: 10.1186/1471-2288-14-122.

7. How were metrological factors controlled in the models? The non-linear associations should be considered by natural splines or DLNM cross-basis patterns.

Response: We added the presentation of the used model (in R statistical software). Meteorological factors (ambient temperature and relative humidity) are presented as natural spline with three degrees of freedom. This study belongs to more general approach under assumption “Ambient air pollution may be damaging every organ and virtually every cell in the human body”. We still fully don’t know which health problems are related to ambient air pollution concentration levels. As the applied methodology allows relatively fast “scan” of the associations we mainly did this. Our findings are supported that for the subgroup (G40-G47-episodic and paroxysmal disorders) we observe more positive statistically significant associations (161 in total) versus more broad G00-G99-nervous system diseases, with 115 positive statistically significant associations. What was proposed (say distributed lag non-linear model or DLNM) can be sued to the detailed study, say migraine attacks.

8. What are the missing rates for air pollutants on both case/control days?

Response: We used the case-crossover (CC) design but didn’t realize it as case/control, rather daily count of events on the clusters. Such approach is (purely) equivalent to the CC method (described in details in Armstrong et al. 2014).

9. Line 157-158, suggest to change this expression. Also, if hundreds of models are performed, multiple testing is needed for type-I error.

Response: The numerical calculations were done in R statistical software using the package GNM with the option “quasipoisson”. The quasi-Poisson regression is realized to model an over dispersed count variable.

As in raised point 7, we realize “to scan” of the potential associations. The number of the correlations increase (from 115 to 161 positive statistically significant associations) if we used ED visits (G40-G47-episodic and paroxysmal disorders) with more health conditions already linked with ambient air pollution (assuming migraine attacks-39% as majority of cases). Also we applied the “quasi-Poisson” option which gives wider 95% confidence intervals. We listed all results, the proposed coloured heat map, which allows to identify the patterns of the associations. These associations can be studied more in details, say ambient air pollution concentration levels and number of ED visits for migraine attacks (from common (without aura) to classical (with aura) to unspecified migraines to complicated migraines/status migrainosus) among females.

In our approach, even the used data from the NACRS are daily counts and time-series type, we don’t have issues of the used degree of freedom for time and/or others often present in GAM methodology.

10. Line 194, it was also found? Here and other grammars/expressions should be checked by peer scientists in this field.

Response: We are not sure we understand the posed question. The manuscript was thoroughly checked by the authors, which are Research Scientist and Principal Investigator (Dr. Szyszkowicz), Study Coordinator (A. Lukina) and Medical Doctor (Dr. Burstein). Additionally, an independent colleague have read the paper and stated that “the paper is well written with no major revisions needed”. However, some smoothing with minor edits were suggested and the paper was corrected accordingly. We hope that this newest version will be satisfactory. 

11. It is not appropriate and offers minimal knowledge by presenting the frequencies in tested associations for each exposure. Reshape this by forest plots or tables that reveals more information.

Response: We have changed the Fig 1 into three additional Figs 1, 2 and 3 that provides fully information for G40-G47. We also expanded our Supplementary Information section by adding three more Figs (S2, S3 and S4)-for G00-G99. We also expanded the results by adding RR and associated 95% CI (lines: 271-275-marked version). ________________________________________

6. PLOS authors have the option to publish the peer review history of their article (what does this mean?). If published, this will include your full peer review and any attached files.

Do you want your identity to be public for this peer review? For information about this choice, including consent withdrawal, please see our Privacy Policy.

Reviewer #1: No

Reviewer #2: No

---

## [Decision Letter · Decision Letter 1]

31 Jan 2022

PONE-D-21-18477R1Urban air pollution and emergency department visits related to central nervous system diseases.PLOS ONE

Dear Dr. Szyszkowicz,

Thank you for submitting your manuscript to PLOS ONE. After careful consideration, we feel that it has merit but does not fully meet PLOS ONE’s publication criteria as it currently stands. Therefore, we invite you to submit a revised version of the manuscript that addresses the points raised during the review process.

We look forward to receiving your revised manuscript.

Kind regards,

Manpreet Singh Bhatti, B.E. (Civil), M.E. (Env. Eng.), Ph.D.

Academic Editor

PLOS ONE

Journal Requirements:

Additional Editor Comments:

Nil

Reviewers' comments:

Reviewer's Responses to Questions

**Comments to the Author**

1. If the authors have adequately addressed your comments raised in a previous round of review and you feel that this manuscript is now acceptable for publication, you may indicate that here to bypass the “Comments to the Author” section, enter your conflict of interest statement in the “Confidential to Editor” section, and submit your "Accept" recommendation.

Reviewer #1: All comments have been addressed

Reviewer #2: All comments have been addressed

Reviewer #3: (No Response)

Reviewer #4: All comments have been addressed

Reviewer #5: (No Response)

2. Is the manuscript technically sound, and do the data support the conclusions?

Reviewer #1: Yes

Reviewer #2: Yes

Reviewer #3: Yes

Reviewer #4: Yes

Reviewer #5: Partly

3. Has the statistical analysis been performed appropriately and rigorously? 

Reviewer #1: Yes

Reviewer #2: Yes

Reviewer #3: No

Reviewer #4: Yes

Reviewer #5: No

4. Have the authors made all data underlying the findings in their manuscript fully available?

Reviewer #1: No

Reviewer #2: Yes

Reviewer #3: No

Reviewer #4: Yes

Reviewer #5: No

5. Is the manuscript presented in an intelligible fashion and written in standard English?

Reviewer #1: Yes

Reviewer #2: Yes

Reviewer #3: Yes

Reviewer #4: Yes

Reviewer #5: Yes

6. Review Comments to the Author

Reviewer #1: (No Response)

Reviewer #2: The authors have responded to my previous comments, but I have to mention that this manuscript still can be polished by editing service.

Reviewer #3: The authors made a sincere effort in addressing the questions raised in the previous round. I have the following additional comments/clarifications:

1. A conditional quasi-Poisson regression model was used for fitting the data. Was there any count of value = 0? In such a situation, zero-augmented models need to be sued.

2. The authors state: "In total, 2,160 statistical models {15 (time lags expressed as days) x 18 (strata) x 8 (air pollutants and air quality health indices)} were applied". If so many models were run, how was the best-fitting model determined (via model comparison statistics, such as AIC/BIC, etc)?

Reviewer #4: The authors have responded well to the statistical issues raised in the previous review. There is no further statistical concern about this revised manuscript.

Reviewer #5: 1. The literature on previous studies was not properly surveyed. A thorough review is recommended.

2. In line number 160, the authors used the formula for generating AQHI values involving an exponential function. Would the authors justify the basis for taking exponential function? Is it possible to generate for AQHI values by using a logarithmic function?

3. The authors should clearly explain the Generalized Non-linear Model. Is it possible to replace the ‘Quasi Poisson’ distribution by any other discrete or continuous distribution?

4. The results of the statistical analysis (Table 3, Table S1, and Table S2) were not represented diagrammatically. The authors did not clearly explain the descriptive statistics. What could be concluded about the data from mean, median, and quartile values.

5. The novelty of the proposed approach has not been highlighted. Moreover, the authors did not emphasize on the significance of the proposed approach over the approaches discussed in the previous studies.

6. It would be better if the authors represent total frequencies (as in Figures 1, 2, 3, S1, S2, and S3) by a frequency diagram.

7. PLOS authors have the option to publish the peer review history of their article (what does this mean?). If published, this will include your full peer review and any attached files.

Reviewer #1: No

Reviewer #2: No

Reviewer #3: No

Reviewer #4: No

Reviewer #5: No

---

## [Author Response · Author response to Decision Letter 1]

7 Mar 2022

Response to Reviews:

Journal Requirements:

Comment: Please review your reference list to ensure that it is complete and correct. If you have cited papers that have been retracted, please include the rationale for doing so in the manuscript text, or remove these references and replace them with relevant current references. Any changes to the reference list should be mentioned in the rebuttal letter that accompanies your revised manuscript. If you need to cite a retracted article, indicate the article’s retracted status in the References list and also include a citation and full reference for the retraction notice.

Response: We have made some significant changes to the Reference list mostly by moving some of the references that already existed in the previous version of the paper but for more consistent flow and also by adding a few additional references as requested by one of the reviewers. References that have been moved are: #17 (old #52): Chen H, Kwong JC, Copes R. et al. Exposure to ambient air pollution and the incidence of dementia: a population-based cohort study. Environ. Int. 2017; 108: 271-277, #18 (old #53): Clifford A, Lang L, Chen R. et al. Exposure to air pollution and cognitive functioning across the life course – a systematic literature review. Environ. Res. 2016; 147: 383-398, and # 34 (old #51): Vicedo-Cabrera AM, Sera F, Liu C. et al. Short term association between ozone and mortality: global two stage time series study in 406 locations in 20 countries. BMJ 2020, 368: m108. doi:10.1136/bmj.m108.), which made many references to shift in their consecutive numbers as well. New references that have been added are #54: Hu C, Fang Y, Li F, Dong B, Hua X, Jiang W, Zhang H, Lyu Y, Zhang X. Association between ambient air pollution and Parkinson’s disease: systematic review and meta-analysis. Environ. Res. 2019; 168: 448-459, #55: Lavery AM, Waubant E, Casper TC, Roalstad S, Candee M, Rose J, Belman A, Weinstock-Guttman B, Aaen G, Tillema J, Rodriguez M, Ness J, Harris Y, Graves J, Krupp L, Charvet L, Benson L, Gorman M, Moodley M, Rensel M, Goyal M, Mar S, Chitnis T, Schreiner T, Lotze T, Greenberg B, Kahn I, Rubin J, Waldman AT. Urban air quality and associations with pediatric multiple sclerosis. Ann. Clin. Transl. Neur. 2018; 5(10): 1146-1153, #56: Abbaszadeh S, Tabary M, Aryannejad A, Abolhasani R, Araghi F, Khaheshi I, Azimi A. Air pollution and multiple sclerosis: a comprehensive review. Neurol. Sci. 2021; 42: 4063-4072, #57: Thilakaratne RA, Malig BJ, Basu R. Examining the relationship between ambient carbon monoxide, nitrogen dioxide, and mental health-related emergency department visits in California, USA. Sci. Total Environ. 2020; 746: 140915, #58: Szyszkowicz M, Thompson EM, Colman I, Rowe BH. Ambient air pollution exposure and emergency department visits for substance abuse. PLoS ONE 2018; 13(6): e0199826, #59: Goebel U, Wollborn J. Carbon monoxide in intensive care medicine-time to start the therapeutic application. Intens. Care Med. Exp. 2020; 8:2, #60: Szyszkowicz M, Kousha T, Kingsbury M, Colman I. Air pollution and emergency department visits for depression: a multicity case-crossover study. Environ. Health Insights 2016; 10: 155-161, and #61: Lin G, Li L, Song Y, Zhou Y, Shen S, Ou C. The impact of ambient air pollution on suicide mortality: a case-crossover study in Guangzhou, China. Environ. Health 2016; 15:90.). To accompany all these new references, the Özkaynak H, Baxter LK, Burke J. Evaluation and application of alternative air pollution exposure metrics in air pollution epidemiology studies. J. Expo. Sci. Env. Epid. 2013; 23: 565 reference have been shifted to its last position as #62. 

We also removed a few references, including: Kim CS, Hu SC. Regional deposition of inhaled particles in human lungs: comparison between men and women. Appl. Physiol. 1998; 84(6): 1834-1844. (#51) and Lipton RB, Mazer C, Newman LC, Solomon S. Sumatriptan relies migraine like headaches associated with carbon monoxide. Headache 1997; 37: 392-395. (#53)

Additional Editor Comments:

Nil

Reviewers' comments:

Reviewer's Responses to Questions

Comments to the Author

1. If the authors have adequately addressed your comments raised in a previous round of review and you feel that this manuscript is now acceptable for publication, you may indicate that here to bypass the “Comments to the Author” section, enter your conflict of interest statement in the “Confidential to Editor” section, and submit your "Accept" recommendation.

Reviewer #1: All comments have been addressed

Reviewer #2: All comments have been addressed

Reviewer #3: (No Response)

Reviewer #4: All comments have been addressed

Reviewer #5: (No Response)

2. Is the manuscript technically sound, and do the data support the conclusions?

Reviewer #1: Yes

Reviewer #2: Yes

Reviewer #3: Yes

Reviewer #4: Yes

Reviewer #5: Partly

3. Has the statistical analysis been performed appropriately and rigorously? 

Reviewer #1: Yes

Reviewer #2: Yes

Reviewer #3: No

Reviewer #4: Yes

Reviewer #5: No

4. Have the authors made all data underlying the findings in their manuscript fully available?

Reviewer #1: No

Reviewer #2: Yes

Reviewer #3: No

Reviewer #4: Yes

Reviewer #5: No

5. Is the manuscript presented in an intelligible fashion and written in standard English?

Reviewer #1: Yes

Reviewer #2: Yes

Reviewer #3: Yes

Reviewer #4: Yes

Reviewer #5: Yes

6. Review Comments to the Author

Reviewer #1: (No Response)

Reviewer #2: The authors have responded to my previous comments, but I have to mention that this manuscript still can be polished by editing service.

Response: We have thoroughly gone throughout the manuscript and corrected/improved sentences (all changes tracked). We hope that this new version is finally to your satisfaction. 

Reviewer #3: The authors made a sincere effort in addressing the questions raised in the previous round. I have the following additional comments/clarifications:

Comment 1. A conditional quasi-Poisson regression model was used for fitting the data. Was there any count of value = 0? In such a situation, zero-augmented models need to be sued.

Response: Thank you for this question. Count value = 0 does not create the problems in the used method. We have 1008 Levels: 2004:1:1, 2004:1:2, …, 2015:12:7 – where each “year:month:day of week” may have 4 or 5 days. It may happen that for a specific stratum we have all days with 0 counts. In the used approach, it is not the problem. This situation creates problem in the standard case-crossover method, where the Cox proportional-hazards model is applied (conditional logistic regression). Before applying such models, we need to remove each stratum with all zeros, but not in here realized method.

Comment 2. The authors state: "In total, 2,160 statistical models {15 (time lags expressed as days) x 18 (strata) x 8 (air pollutants and air quality health indices)} were applied". If so many models were run, how was the best-fitting model determined (via model comparison statistics, such as AIC/BIC, etc)?

Response: We organized our data as time-series daily counts (and daily environmental factors). The model was fitted to the triple combination (i.e., stratum, air pollutant (five individual pollutants and two air quality health indices), and time lag expressed as days); therefore, we have such 2,160 statistical combinations of the triples. We do not validate (test AIC/BIC) the constructed model. In the used approach we do not have “time” as it is used in the GAM (GLM), where we need to model time variable (and validate the fitted models). Here, the constructed clusters controls for “time” – we conditioned to the sum of counts for each cluster. We added a mathematical description (lines 214-221-marked version).

Reviewer #4: The authors have responded well to the statistical issues raised in the previous review. There is no further statistical concern about this revised manuscript.

Response: Thank you for reviewing our manuscript and providing constructive comments. 

Reviewer #5: Comment 1. The literature on previous studies was not properly surveyed. A thorough review is recommended.

Response: We have expanded on the discussion section and added more information on health effects associated with CO exposure and other ambient air pollutants (fine particulate matter and sulfur dioxide), the literature that was included is recent starting from 2016 (Szyszkowicz M, Kousha T, Kingsbury M, Colman I. Air pollution and emergency department visits for depression: a multicity case-crossover study. Environ. Health Insights 2016; 10: 155-161 and Lin G, Li L, Song Y, Zhou Y, Shen S, Ou C. The impact of ambient air pollution on suicide mortality: a case-crossover study in Guangzhou, China. Environ. Health 2016; 15:90.) and ending with the most recent from 2020 and 2021 (Abbaszadeh S, Tabary M, Aryannejad A, Abolhasani R, Araghi F, Khaheshi I, Azimi A. Air pollution and multiple sclerosis: a comprehensive review. Neurol. Sci. 2021; 42: 4063-4072, Thilakaratne RA, Malig BJ, Basu R. Examining the relationship between ambient carbon monoxide, nitrogen dioxide, and mental health-related emergency department visits in California, USA. Sci. Total Environ. 2020; 746: 140915., and Goebel U, Wollborn J. Carbon monoxide in intensive care medicine-time to start the therapeutic application. Intens. Care Med. Exp. 2020; 8:2.) (lines: 333-350-marked version).

Comment 2. In line number 160, the authors used the formula for generating AQHI values involving an exponential function. Would the authors justify the basis for taking exponential function? Is it possible to generate for AQHI values by using a logarithmic function?

Response: Thank you for this question. The air quality health index (or simply AQHI) is calculated as specified in the cited publication (Stieb DM et al, A new multipollutant, no-threshold air quality health index based on short-term associations observed in daily time-series analyses. J. Air Waste Manag. Assoc. 2008; 58(3):435-450. doi:10.3155/1047-3289.58.3.435). Since 2008, the AQHI is a standard scale in Canada designed for general Canadian population to understand how air quality means to their health. It is scaled as 1-10 or 10+ and is hourly reported to Canadian population. It is a preventive tool – the higher the number, the greater the health risk associated with the air quality, meaning that persons are advised to limit their outdoor activities (i.e., 1-3 mean “low health risk”, 4-6 mean “moderate health risk”, 7-10 mean “high health risk”, and 10+ mean “very high health risk”). In summary – we have not changed the definition (exp(x) vs. log(x)). It is an interesting proposition to apply “log”. We cannot modify the representation – as it will no longer be AQHI, which is an approved/accepted scale in Canada.

Comment 3. The authors should clearly explain the Generalized Non-linear Model. Is it possible to replace the ‘Quasi Poisson’ distribution by any other discrete or continuous distribution?

Response: Thank you for this question. There are a few methods, which can be used – main categories of the methods are: time-series (TS) or case-crossover (CC) designs.

In TS, daily counts are analyzed, usually generalized linear model (GLM) is used and Quasi-Poisson is applied as a link function.

In the standard CC method, individual events and matched controls period are analyzed. The conditional logistic regression is usually applied.

In the present study, we analyze daily counts (as in TS), but we are using the time-stratified CC design to control for “time” variable. The applied method and comparison with other techniques (+ software) are presented in Armstrong et al. Conditional Poisson models: a flexible alternative to conditional logistic case cross-over analysis. BMC Med. Res. Methodol. 2014; 14:122. doi: 10.1186/1471-2288-14-122. 

We added some mathematical descriptions. The methodology is well represented/described in the above-cited reference.

Comment 4. The results of the statistical analysis (Table 3, Table S1, and Table S2) were not represented diagrammatically. The authors did not clearly explain the descriptive statistics. What could be concluded about the data from mean, median, and quartile values.

Response: The descriptive statistics for air pollutants used in our study are described on lines 242-253-marked version), mean, maximum and minimum levels are presented for each air pollutant (and air health index) for two seasons (warm and cold) and year round. As for supplemental materials, the ED visits (as daily counts) and environmental characteristics data are mainly for later publications and for other centers to compare/reference– we characterized the environmental conditions in Toronto, Canada. In addition, on the e-location already provided in the manuscript (https://github.com/szyszkowiczm/NERVEToronto), which contains the histograms of all used air pollutants, weather factors and the map of Toronto (population density and monitoring stations).

The results are generated in 3D space with the coordinates (stratum, air pollutant, lag – 3 axes). The figures S1, S2, and S3 are the projections on 2D – show scores of the significant results.

Comment 5. The novelty of the proposed approach has not been highlighted. Moreover, the authors did not emphasize on the significance of the proposed approach over the approaches discussed in the previous studies.

Response: We emphasised the uniqueness of the present study in introduction (lines 119-129-marked version); however, we added a bit more details of our study compared to previous studies (lines 123-125-marked version). 

Comment 6. It would be better if the authors represent total frequencies (as in Figures 1, 2, 3, S1, S2, and S3) by a frequency diagram.

Response: We tabulated all the results from 2,160 models (RR and 95%CIs), also we included histogram/frequencies of the environmental factors and the map of Toronto in e-location mentioned on 227-marked version. Figure S4 and S5 are maps to the numerical results (RR, 95%CIs) for G00-G99 “all nervous system diseases” and G40-G47 “episodic and paroxysmal disorders”, respectively.

7. PLOS authors have the option to publish the peer review history of their article (what does this mean?). If published, this will include your full peer review and any attached files.

Do you want your identity to be public for this peer review? For information about this choice, including consent withdrawal, please see our Privacy Policy.

Reviewer #1: No

Reviewer #2: No

Reviewer #3: No

Reviewer #4: No

Reviewer #5: No

---

## [Decision Letter · Decision Letter 2]

12 Jun 2022

Urban air pollution and emergency department visits related to central nervous system diseases.

PONE-D-21-18477R2

Dear Dr. Szyszkowicz,

We’re pleased to inform you that your manuscript has been judged scientifically suitable for publication and will be formally accepted for publication once it meets all outstanding technical requirements.

Kind regards,

Julian Aherne

Academic Editor

PLOS ONE

Additional Editor Comments (optional):

All three reviewers' recommend 'accept' as all comments have been addressed. Well done. I look forward to seeing your published manuscript.

Reviewers' comments:

Reviewer's Responses to Questions

**Comments to the Author**

1. If the authors have adequately addressed your comments raised in a previous round of review and you feel that this manuscript is now acceptable for publication, you may indicate that here to bypass the “Comments to the Author” section, enter your conflict of interest statement in the “Confidential to Editor” section, and submit your "Accept" recommendation.

Reviewer #2: All comments have been addressed

Reviewer #3: All comments have been addressed

Reviewer #4: All comments have been addressed

2. Is the manuscript technically sound, and do the data support the conclusions?

Reviewer #2: Yes

Reviewer #3: (No Response)

Reviewer #4: Yes

3. Has the statistical analysis been performed appropriately and rigorously? 

Reviewer #2: Yes

Reviewer #3: (No Response)

Reviewer #4: Yes

4. Have the authors made all data underlying the findings in their manuscript fully available?

Reviewer #2: No

Reviewer #3: (No Response)

Reviewer #4: Yes

5. Is the manuscript presented in an intelligible fashion and written in standard English?

Reviewer #2: Yes

Reviewer #3: (No Response)

Reviewer #4: Yes

6. Review Comments to the Author

Reviewer #2: The authros have responded well to all my previous comments in R1. While the new text is appropriate in content, there are multiple problems with English syntax. Copyediting is required.

Reviewer #3: (No Response)

Reviewer #4: The authors have responded well to the statistical issues raised in the previous review. There is no further statistical concern about this revised manuscript.

7. PLOS authors have the option to publish the peer review history of their article (what does this mean?). If published, this will include your full peer review and any attached files.

Reviewer #2: No

Reviewer #3: No

Reviewer #4: No

---

## [Editor Report · Acceptance letter]

17 Jun 2022

PONE-D-21-18477R2 

Urban air pollution and emergency department visits related to central nervous system diseases. 

Dear Dr. Szyszkowicz:

I'm pleased to inform you that your manuscript has been deemed suitable for publication in PLOS ONE. Congratulations! Your manuscript is now with our production department. 

Kind regards, 

on behalf of

Dr. Julian Aherne 

Academic Editor

PLOS ONE